# Stock Assessment of the Commercial Small Pelagic Fishes in the Beibu Gulf, the South China Sea, 2006–2020

**DOI:** 10.3390/biology13040226

**Published:** 2024-03-29

**Authors:** Xiaofan Hong, Kui Zhang, Jiajun Li, Youwei Xu, Mingshuai Sun, Shannan Xu, Yancong Cai, Yongsong Qiu, Zuozhi Chen

**Affiliations:** 1College of Marine Sciences, Shanghai Ocean University, Shanghai 201306, China; jyhdhkhxf114209@gmail.com; 2South China Sea Fisheries Research Institute, Chinese Academy of Fishery Sciences, Guangzhou 510300, China; zhangkui@scsfri.ac.cn (K.Z.); lijiajun@scsfri.ac.cn (J.L.); xuyouwei@scsfri.ac.cn (Y.X.); wasersun@163.com (M.S.); xushannan@scsfri.ac.cn (S.X.); onion-20062006@163.com (Y.C.); qys@scsfri.ac.cn (Y.Q.); 3Key Laboratory for Sustainable Utilization of Open-Sea Fishery, Ministry of Agriculture and Rural Affairs, Guangzhou 510300, China

**Keywords:** commercial small pelagic fishes, length frequency analysis, population dynamics, Beibu Gulf, coastal fishery

## Abstract

**Highlights:**

**What are the main findings?**
The commercial small pelagic fishes (Decapterus maruadsi and Trachurus japonicus) in the Beibu Gulf were still miniaturizing.Fisheries management, characterized by reduced fishing efforts, cannot completely restore population structure in a short period.

**What are the suggestions for the future fisheries development in the Beibu Gulf?**
Continuing to maintain low fishing mortality and increasing the catchable length should be the key ways to achieve fishery resource conservation and recovery.

**Simple Summary:**

*Decapterus maruadsi* and *Trachurus japonicus*, as the main commercial small pelagic fish in the coastal fisheries of China, have been facing the threat of population decline due to overfishing. In this work, we assessed the population status of two commercial small pelagic fish stocks in the Beibu Gulf (in the South China Sea) over the past 15 years (2006–2020). The analysis results show that the commercial small pelagic fishes in the Beibu Gulf were still miniaturizing, and fisheries management characterized by reduced fishing efforts cannot completely restore population structure in a short period. Continuing to maintain low fishing mortality and increasing the catchable length should be the key ways to achieve fishery resource conservation and recovery. Our findings will provide a key scientific basis for future improvements in offshore fisheries management.

**Abstract:**

Long-term variations in population structure, growth, mortality, exploitation rate, and recruitment pattern of two major commercial small pelagic fishes (CSPFs) (*Decapterus maruadsi* and *Trachurus japonicus*) are reported based on bottom trawl survey data collected during 2006–2020 in the Beibu Gulf, South China Sea. All individuals collected during each sampling quarter over a period of 15 years were subjected to laboratory-based analysis. In this study, the stock of *D. maruadsi* and *T. japonicus* inhabiting the Beibu Gulf was assessed using length-based methods (bootstrapped electronic length frequency analysis (ELEFAN)) to complete stock assessment in different fishery management periods (the division of fisheries management periods was based on China’s input and output in the South China Sea offshore fisheries over 15 years, specifically divided into period I (2006–2010), period II (2011–2015), and period III (2016–2020)). The results showed that the mean body length, dominant body size, and estimated asymptotic length of two CSPFs decreased, whereas their growth coefficient decreased, indicating miniaturization and slower growth, respectively. Estimated exploitation rates and catching body length for two CSPFs indicated that both stocks in the Beibu Gulf were overexploited in period I and moderately exploited after 2011. These stocks were taking a good turn in status in period III, with the exploitation rate much lower than the initial period and reversing the downward trend in catching body length. Furthermore, the variations in the spawning season of the two CSPF stocks and their barely satisfactory expected yield indicated the complexity of the current fishery management in the Beibu Gulf. These results suggest that management measures to reduce fishing pressure may have a positive influence on the biological characteristics of those CSPFs in the Beibu Gulf; however, the stock structure already affected by overfishing will be a huge challenge for the conservation and restoration of fisheries resources in the future. Given that the current stocks of *D. maruadsi* and *T. japonicus* in the Beibu Gulf still have low first-capture body length (*L*_c_) and high fishing mortality (*F*) (compared to *F*_0.1_), we identify a need to refine population structure by controlling fishing efforts and increasing catchable size, and more consideration should be given to the local fishery resource status in fisheries management.

## 1. Introduction

Fish and other aquatic food products provide more than 15% of animal protein to a third of the planet’s population and are important sources of essential micronutrients [1]. Although coastal fisheries only account for 3% of the ocean surface, their total annual catch is currently between 50 and 60 million tons, approximately half of the global marine catch [2]. There is evidence that catches are still insufficient in developed or rapidly developing countries due to the growing popularity of seafood [3]. The rapid development of coastal fisheries in China began in the late 1970s [4]. Its development has not only significantly increased the income of rural populations but also contributed more significantly to local and national economies [5]. However, as the world’s largest producer of marine fishery [6], China has experienced a significant decline in the quality and quantity of its coastal fishery resources in the past few decades due to the lack of restrictions on fishing yields [7,8], which makes it urgent to promote the sustainable development of fishery resources and the restoration of marine ecosystem functions. Over the past two decades, the Chinese government has promulgated and implemented a series of fishery management policies and measures, among which the most far-reaching ones are the “summer fishing moratorium” and “‘Zero-growth’ and ‘Negative-growth’ strategies” [9,10]. The specific contents can be summarized as suspending fishing operations during the fishing moratorium, limiting national total fisheries production, and setting targets for negative growth in fishing capacity, respectively. Although these measures have been effective, frequent human activities (e.g., marine fishing or pollution) and climate change will inevitably have a strong impact on coastal fishery ecosystems due to ongoing environmental changes, affecting fish communities and subsequently affecting fishery yields [11]. Therefore, it is necessary to evaluate the effectiveness of past fishery management, which is crucial for the development of management measures that are more suitable to the current status of local fishery resources [12].

*Decapterus maruadsi* and *Trachurus japonicus* were distributed on the continental shelf waters in the sublittoral zone and may enter semi-enclosed sea areas [13,14]. They were both major commercial pelagic small fish species in the Beibu Gulf (in the northern South China Sea), and also the dominant catch in this fishing ground [15]. Despite being a traditional fishing ground with high fishery yields in China, previous research has shown that the fishery ecosystem in the Beibu Gulf has significantly degraded due to overfishing [16]. The biomass and abundance of fishing targets, including *D. maruadsi* and *T. japonicus*, have decreased significantly [17]. Obviously, previous fishery management policies have not effectively prevented the decline of fishery resources in the region as a whole. Fisheries management issues are inherently multidisciplinary; good and sustainable fisheries management must achieve the right balance between the effectiveness of its regime and its cost of design, implementation, and operation [18]. Anthropogenic factors (especially fishing operations) and climate change can cause significant changes in the dynamics of fish populations by altering growth, mortality, maturity, and recruitment at the population level [19,20]. Undoubtedly, the most direct and effective method to evaluate the effectiveness of fisheries management is through minorizing and assessing the population dynamics of the fishing targets [21]. Therefore, fishery stock assessments are essential for science-based fisheries management, and their accuracy has always been a focus of fisheries resource research [22].

The development and utilization of fisheries stock assessment models are fundamental to implementing modern management and maintaining the sustainability of fisheries [23]. These models are essentially demographic analyses designed to determine the effects of fishing on fish populations and to evaluate the potential consequences of alternative management policies [24]. The conceptual framework for traditional assessment models is based on a large amount of demographic process data, including birth, mortality, maturation, age structure, and so on. However, only 10–50% of stocks were assessed in developed countries compared with 5–20% in developing countries; fully accessing those data is generally difficult to achieve in expanding fisheries, especially in the fisheries of developing countries [25]. Studies have shown that where fisheries are intensively managed, stocks are above target levels or rebuilding, whereas where fisheries management is less intense, stock status and trends are worse [26]. As a result, there is a growing recognition of the negative impact of poor data situations on fisheries management, and an increasing number of emerging stock assessment models have been developed and widely applied to address this issue [27]. The determination of rates of body growth is the first step in many aquatic population studies and fisheries stock assessments [28]. Electronic length frequency analysis (ELEFAN) is a widely used method to fit a von Bertalanffy growth curve to length frequency distribution data for data-poor fisheries [29], which has been performed well in the assessment of fishery resources in China’s coastal waters, e.g., Yellow Sea, South China Sea [23,30]. In order to improve the accuracy and reduce the uncertainty of this stock assessment model, novel calibration and testing methods have been continuously introduced to enhance the model [28,31,32].

Although previous studies have extensively examined the commercial fish stocks in the Beibu Gulf [15,33], evaluation of the effectiveness of fisheries management over large timescales is still rare. In this study, we applied the above-mentioned length-based methods (bootstrapped ELEFAN) to stock assessment for two exploited commercial small pelagic fish (CSPF), *D. maruadsi* and *T. japonicus*, in the Beibu Gulf. The objectives of this study were to: (1) provide an overview of the population status of exploited CSPF stocks in the Beibu Gulf; (2) analyze the effects of past fishery management on the conservation and maintenance of CSPF stocks in the Beibu Gulf; and (3) provide science-based recommendations for improving future fishery management policies and measures in the Beibu Gulf. Overall, the results of this study may contribute to providing a scientific basis to assist in the sustainable utilization and management of fish stocks in data-poor situations.

## 2. Materials and Methods

### 2.1. Study Area

The study area is located northwest of the South China Sea and is surrounded by China and Vietnam on three sides. It is a typical subtropical semi-enclosed bay with an area of about 130,000 km^2^ and a water depth ranging from 5–100 m. Due to the Beibu Gulf being an important traditional fishing ground for China and Vietnam, the area has been regularly monitored over the past 20 years under standardized fisheries resource surveys. As part of the northern South China Sea monitoring program for fisheries resources since 2000, the South China Sea Fisheries Research Institute, Chinese Academy of Fishery Sciences, conducts specific seasonal bottom trawl surveys in this area at least twice a year.

### 2.2. Sampling

Fish sampling data were obtained from 34 bottom trawl surveys at 52 stations between 2006 and 2020, with less than 2 surveys conducted annually (Figure 1). Most survey cruises did not cover all 52 stations for a variety of reasons, but the samples were generally coherent in methodology. The surveys were undertaken by a commercial fishing vessel. In every survey, each station was trawled once for 1 h with a towing speed of 2.5–3.5 knots. Every trawl sample was sorted onboard and identified to the species level. Samples of each fish were counted, weighed, and recorded, and length and weight frequency data were also collected for major commercial fish. When fewer than 50 individuals were caught in a trap, all individuals were measured; otherwise, 50 individuals were randomly sampled for measurement. For each marine fish, body length (mm) and body weight (g) were recorded.

### 2.3. Data Analysis

According to the statistical data from the China Fishery Statistical Yearbook (2006–2020), based on China’s marine fishery yields in the South China Sea and the fishing efforts invested in the region, the study timeline was divided into three periods, which represent different stages of the development of fishery resources in the South China Sea (Figure 2). The specific characteristics of the three fishery management periods can be described as follows: period I (2006–2010)—this is a stage where fishery investment continues to increase significantly while fishery production remains relatively stable; period II (2011–2015)—this is a stage where fishery investment fluctuates and increases, and fishery production continues to rise; period III (2016–2020)—this is a stage where fishery investment shows a basic downward trend and fishery production decreases significantly. The length frequency data of *D. maruadsi* and *T. japonicus* were both processed separately according to the three periods as divided above.

#### 2.3.1. Length–Weight Relationship of Fish Samples

The power relationship between fish body length and weight is described by the following equation:$$W=aL^{b}$$
where *W* represents body weight (g), *L* represents standard length (cm), *a* is a constant condition factor, and *b* is an allometric growth parameter.

#### 2.3.2. Stock Assessment by ELEFAN

Length frequency data of CSPFs from 2006 to 2020 in the Beibu Gulf were pooled and converted to monthly catches with the assumption that the samples were representative of the total catch of the month [34]. We used the function “ELEFAN_GA_boot” in the R package “fishboot” [28] to estimate the von Bertalanffy growth function (VBGF) parameters of CSPFs. “ELEFAN_GA_boot” performs a bootstrapped fitting of VBGF via the function “ELEFAN_GA”, and the latter is based on a genetic algorithm (“G.A.”), a metaheuristic inspired by the process of natural selection [35,36]. The full bootstrap simulations in “fishboot” were used to determine the variability in the search algorithm as well as the uncertainty in sampling the fish [28], and we carried out 200 simulations for each ELEFAN_GA algorithm and obtained statistics for each model’s parameters.

Considering that fish growth varies throughout the year due to the influence of internal and external factors, this study modeled the inter-annual variability in growth for the two CSPFs via a seasonal oscillation VBGF (soVBGF) [37,38]:(1)$$L_{t}=L_{inf}\left(1-e^{-\left(K\left(t-t_{0}\right)+S\left(t\right)-S\left(t_{0}\right)\right)}\right)$$
(2)$$S\left(t\right)=\frac{CK}{2\pi }\sin\left(2\pi \left(t-t_{s}\right)\right)$$
(3)$$S\left(t_{0}\right)=\frac{CK}{2\pi }\sin\left(2\pi \left(t_{0}-t_{s}\right)\right)$$
where *L_inf_* is the asymptotic length, *L_t_* is the body length at age *t*, *K* is the growth coefficient (also referred to as the growth constant), *t*_0_ is the theoretical age at zero length, *C* is a constant (between 0 and 1) indicating the magnitude of the oscillation, and *t_s_* (between 0 and 1) defines the beginning of the (positive) sine wave. The value of *t*_0_ was estimated using the empirical equation [39] as follows:(4)$$\log_{10}\left(t_{0}\right)=-0.3922-0.2751\log_{10}\left(L_{\infty }\right)-1.038\log_{10}K$$

To compare growth parameters among different sampling periods and different studies, a growth performance index (phi-prime index, *φ*′) [40] was calculated as follows:(5)$$\varphi^{\prime }=\log_{10}K+2\log_{10}\left(L_{\infty }\right)$$

The instantaneous total mortality rate (*Z*) was computed from the length frequency data based on the linearized length-converted catch curve [41]. Natural mortality (*M*) was estimated by applying an updated version of the [42] growth-based method [43]:(6)$$M=4.118K^{0.73}{L_{\infty }}^{-0.33}$$

Based on the relationship between *Z* and *M*, fishing mortality (*F*) and exploitation ratio (*E*) were specifically calculated as follows:(7)F=Z−M
(8)E=F/Z

The initial settings in the ELEFAN functions can affect the estimates [44]. To improve the precision of the growth parameters (*L_∞_* and *K*) estimation, the length frequency data of CSPFs was binned according to the maximum body length (*L_max_*) observed for the fish species [45], and the moving average (MA) was set to 9 [44].

Furthermore, a modification of Pope’s virtual population analysis (VPA) and Jones’s length-converted cohort analysis were used to reveal the stock size and composition [32]. The length-based yield per recruit model by Beverton and Holt [46] was used to evaluate the exploitation levels of the two CSPFs while outputting biological reference points such as *F*_max_ (the mortality level when the yield is at a maximum) [47] and *F*_0.1_ (the fishing mortality rate where the slope of the yield per recruit curve is 10% of its value at the origin) [48] to predict optimum yield. This model builds on the output of the length-based VPA [49].

## 3. Results

### 3.1. Size Distributions and Length–Weight Relationship of the Stocks

In this study, the body length frequency of *D. maruadsi* and *T. japonicus* in the three periods was analyzed at 1.5 cm standard length intervals (Figure 3). For most periods, the median and mean of the sampled individuals’ body sizes of the two pelagic fishes were basically the same (Table 1). The mean size of the two pelagic fishes generally decreased over time.

The condition factor and allometric growth parameter in the length–weight relationship varied over time for *D. maruadsi* and *T. japonicus* (Figure 4). Over the three periods, condition factor values of both CPFs showed an increased trend, whereas allometric growth parameter values showed the opposite trend.

### 3.2. Growth Parameters and Growth Curves

The VBGF parameters for the two CPFs changed from 2006–2020 (Table 2). The asymptotic length (*L*_∞_) of *D. maruadsi* had decreased to 26.19 cm in periods II since the maximum (30.43 cm) in period I, and then increased to 26.48 cm in period III; however, the *L*_∞_ of *T. japonicus* had slightly increased to 26.13 cm in period II compared to the period I (26.08 cm) and then decreased to 24.22 cm in period III. The K of both CPFs was continuously decreased over time. The growth parameter index of *D. maruadsi* varied between 2.43 and 2.87, and the growth parameter index of *T. japonicus* varied between 2.58 and 2.66.

Based on the growth parameters (*L*_∞_, *K*, *C*, and *t*_s_) obtained using the “ELEFAN_GA_boot” method and the calculation formula for *t*_0_, the growth curves fitted by the seasonally oscillating VBGF display significant differences in the growth status of *D. maruadsi* and *T. japonicus* during different periods, especially with a noticeable downward trend in the curve for period III compared to period I (Figure 5).

### 3.3. Mortality and Selectivity

The instantaneous total mortality (*Z*) and instantaneous fishing mortality (*F*) of the two CPFs were significantly smaller in the third and second periods compared to the first period, whereas their natural mortality (*M*) did not change significantly in the three periods (Table 3). The exploitation rate (*E*) of *D. maruadsi* and *T. japonicus* ranged from 0.42–0.74 and 0.55–0.80, respectively. Moreover, the values of *L*_50_ and *L*_75_ for both *D. maruadsi* and *T. japonicus* showed the highest magnitude in the first period and the lowest magnitude in the second period.

### 3.4. Stock Status

The phenomenon of catching body length classes of less than 9.75 cm only occurred in the second period for the two CPF populations, and their fishing mortality in each body length class significantly decreased over time, especially in the intermediate and upper body length classes (15.75–24.75 cm). For both CPF populations, compared to the first period, the body length class corresponding to the peak of the mean biomass ratio slightly increased in the second and third periods (Figure 6).

### 3.5. Recruitment and Yield per Recruit

Based on the growth parameters above, the length frequency (LFQ) data can be extrapolated backward onto the time axis to indicate the relative recruitment pattern of stock (Figure 7). The pattern exhibited by the size distribution of the two CSPFs indicated that their major peaks of recruitment occurred in winter and spring in the first and second periods, but their major peaks occurred in April in the third period.

The estimation of the biological reference points for the two CSPFs showed that their optimal exploitation and fishing mortality rates (*E*_max_ and *F*_max_) were both well below the actual values of the current exploitation and fishing mortality in the first period, respectively, while in the third period, neither *F* nor *E* exceeded the Fmax and Emax of their respective stocks, respectively (Figure 8). The body length at first capture (*L*_c_) of two CSPFs exhibited a trend of first decreasing and then increasing during three periods. However, the distribution of current yield per recruit (*YPR*_c_) in *T. japonicus* did not follow the single-grain distribution pattern of *D. maruadsi*, but rather continued to decline over time (Table 4). Only the optimal yield per recruit (*YPR*_max_) of *D. maruadsi* returned to similar levels in the third period as in the initial period, while these two biological reference points of *T. japonicus* also continued to decrease in the three periods. The *F*_0.1_ of the two fishes remains relatively stable during three time periods, and their *YPR*_0.1_ is only slightly lower than the *YPR*_max_ of the same period.

## 4. Discussion

Over the past 15 years, significant variations in population structure, growth, mortality, exploitation rate, and recruitment pattern of two CSPF stocks in the Beibu Gulf have taken place. The mean body length, medium body length, maximum body length, and estimated asymptotic length of these CSPFs in the Beibu Gulf have decreased, indicating a negatively impacted size structure of populations. The values of allometric growth parameter *b* of *D. maruadsi* and *T. japonicus* both showed a downward trend in general and even less than 3.0 in the third period, indicating negative allometric growth of these two CSPFs in the Beibu Gulf [50], however, they were still in the range of 2.5–3.5 of the length–weight relationships of fish [51]. The growth performance index *φ*′ is an indicator of fish growth and allows the comparison of growth performance as represented by the index [52] of stocks of the same or different species, be they closely allied species or not [40], and this index has been widely used in growth studies. The results of the present study showed that the estimated values of *φ’* for *D. maruadsi* have varied more than those of *T. japonicus* over the past 15 years, whereas the growth performance of the former has been slightly better than that of the latter in general. Although the changes in fishing pressure in the past have not significantly weakened the general growth status of fish, there has been a slight decrease in their *φ*′ from 2006 to 2020. It was evident that the recovery of the stocks was still constrained in some ways, due to factors such as salinity [53], temperature [54], stock density [55], etc.

The shrinking body size of fish is an important reflection of the decline in global marine fishery resources, especially in coastal fisheries [56,57]. In the offshore fishing grounds of the South China Sea, the trend towards shrinking commercial fish body size was already a common phenomenon, e.g., threadfin porgy (*Evynnis cardinalis*) [58] and threadfin bream (*Nemipterus virgatus*) [59]. Fish body size is a key biological and ecological trait [60]. The present study showed that *D. maruadsi* and *T. japonicus* in the Beibu Gulf have become miniaturized with a decrease in their growth rate from 2006 to 2020, which is consistent with the common view that fishing can lead to a slowdown in fish growth [61]. The consequences of fishing-induced evaluation (FIE) selective removal may be the main reason for the shrinking body size of both CSPF populations [62]. As fishing mortality usually increases with body size (eventually exceeding the natural mortality of most exploited populations), when faster-growing individuals are removed faster by fishing due to their relatively large size, the surviving population will be dominated by individuals with slower growth [63]. There is a large-scale trawling operation in the fishing grounds of the Beibu Gulf [64], which is usually located in offshore areas with less fishing gear selection [65]. Coupled with the impact of bottom trawling on habitats [66], those negative effects may exert significant fishing pressure on pelagic fishes, forcing them to undergo adaptive evolution [61,67]. Furthermore, the shrinkage in the body size of fish is also related to global warming [68], especially in tropical and intermediate latitudinal areas [69]. Audzijonyte et al. [60] found that temperature indeed drives spatial and temporal changes in fish body size, particularly in small-body fish that tend to shrink with warming. Previous studies have consistently shown that the seawater temperature in the northern South China Sea has been continuously increasing [70]. Climate change undoubtedly has some negative impacts on the recovery of small pelagic fish populations in the Beibu Gulf. Therefore, the FIE caused by past fishing and the shrinking in fish size due to the ongoing effects of climate change appear to be the main drivers of the decline in Beibu Gulf’s CSPF population. Although more and more research and methods have focused on distinguishing fishing effects from environmental effects, these studies still need to be further analyzed in conjunction with more data [71].

Overfishing is the main reason fish populations have declined, as more adaptable groups replace the niche of disappearing groups, leading to changes in fish population structure [72]. In the present study, estimated exploitation rates indicate that these stocks were overexploited in period I (2006–2010), and the level of exploitation has been reduced to varying degrees after 2010. In the third period (2016–2020), the status of these populations improved significantly, with a substantial decrease in exploitation rates compared to the first period. Although the policies and measures adopted by the Chinese government in offshore fishery management have significantly reduced the fishing pressure on pelagic fishes in the Beibu Gulf, *D. maruadsi* and *T. japonicus* were still experiencing excessive fishing pressure (*E* > 0.5) [73]. In the context of previous intense overfishing, the changes in the proportion of biomass within the population triggered by the recovery of captured size and the decrease in fishing mortality rates of different length groups prove without a doubt that the decreasing of fishing pressures is effective for population structure recovery [74]. However, the decline in fishing pressure alone will not result in a rapid or full recovery of pelagic fish stocks [75], and the impact of climate change on the sustainable development of the marine commercial fishing industry is also receiving increasing attention from scientists and policymakers.

In many marine ecosystems, climate change will result in environmental conditions that are beyond the range of variability experienced since the advent of industrialized fishing [76]. Climate change is projected to influence multiple aspects of the pelagic marine environment, including temperature, circulation, chemistry (pH), stratification, productivity, and oxygen availability. In particular, warming, changes in circulation patterns, and altered pelagic food webs due to climate change are likely to drive changes in the distribution, spawning and migration behaviors, early life survival, and recruitment of pelagic fishes [77,78]. The El Nino–Southern Oscillation also affects the catch of *D. maruadsi* and the distribution of its central spawning grounds in Zhejiang coastal waters [79]. Past research suggests that the spawning seasons of *D. maruadsi* and *T. japonicus* were mainly in the spring and summer seasons [79,80]. Although the distribution of the recruitment pattern of both CSPFs in the Beibu Gulf during the three periods was generally consistent with previous studies, the peak of the third period is clearly concentrated in April (Figure 7), indicating a significant compression of the spawning season for fish populations. Fish can avoid higher temperatures through a poleward shift in the spawning area and a temporal shift in spawning timing [81]; however, the geographical constraints of the northern land in the Beibu Gulf hinder the migration of fish in this region, which may lead to a more concentrated spawning season to match the “short window” of time optimal for spawning. Therefore, as global warming continues to increase in the future [82,83], fisheries management policies and measures should also consider the impact of climate change on population structure and recruitment patterns [84]. In this study, the *F*_max_ of the two fishes in the Beibu Gulf increased significantly with the continuous and effective implementation of various fisheries management measures, although during period III, *F*_max_ was higher than the corresponding actual fishing intensity (*F*), but their small population size made it impossible to increase their yield significantly even with increased fishing effort; therefore, *F*_max_ is not suitable as the target reference point here. Therefore, this study suggests that future fishing management in the Beibu Gulf should adopt a more conservative *F*_0.1_ as the target reference point. On the one hand, this can continue to reduce the actual fishing intensity for fishes and protect stock resources; on the other hand, the *YPR*_0.1_ of the two fishes only slightly decreases compared to *YRP*_max_, so the negative impact of reducing fishing pressure on production is minimal. The fishery resources in the Beibu Gulf still need further conservation and restoration. In order to achieve this goal, it is necessary to improve the selectivity of fishing gear for small individuals while maintaining the current (low) level of fishing [12,85]. Additionally, this study suggests that the closed fishing season can be optimized or refined based on different water economic fish species supplement patterns, taking into account the diverse types of offshore fishing grounds in China. Dynamic management is expected to increase the efficacy and efficiency of fisheries management over static approaches by better aligning human and ecological scales of use [86]. In the face of the complex changes in the habitat brought about by global warming in the future [87,88], it is crucial for the sustainable exploitation of offshore fisheries to transform the relatively uniform national management method into fine fishery management based on the actual status of local fishery resources.

## 5. Conclusions

In this study, the stock assessment of pelagic fishes was mainly based on the fishery resources survey data by bottom trawling in Beibu Gulf, and although the average depth of Beibu Gulf is shallow, the catch capacity of bottom trawl gear for pelagic fish may be weaker than that for benthic fish. Therefore, future research work should combine commercial fishing data to achieve a more accurate assessment of the status of pelagic fish stocks in the Beibu Gulf.

The analysis of the population assessment results for *D. maruadsi* and *T. japonicus* in the Beibu Gulf offshore fishery indicates that these two common commercial pelagic fishes have been overexploited in the past 15 years. Although the reduction in fishing pressure has restrained the declining trend of the populations, the recovery of their resources still faces multiple obstacles. This study has demonstrated that reduced fishing pressure has a positive effect on the conservation of small pelagic commercial fish stocks in the Beibu Gulf; however, further studies should consider other mechanisms, such as the direct or indirect effects of climate change-related environmental factors on the population status in this region. A more comprehensive approach, including multispecies and ecosystem analyses, will enable the development of fisheries management strategies that are more aligned with the local fisheries resource status.

## Figures and Tables

**Figure 1 biology-13-00226-f001:**
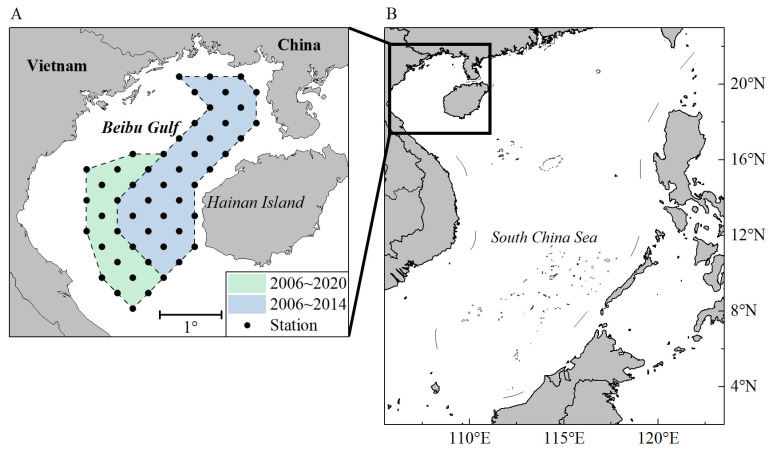
The sampling stations of the waters in the Beibu Gulf during 2006–2020. (**A**) Sampling regions (blue and green) and sampling stations (dots) in the Beibu Gulf; (**B**) the location of the Beibu Gulf in the South China Sea.

**Figure 2 biology-13-00226-f002:**
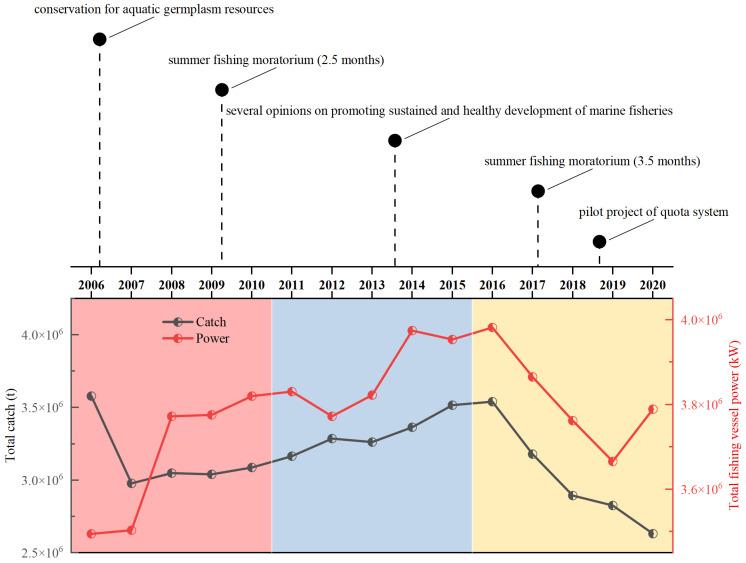
Offshore marine fishery efforts in the South China Sea and timeline of the introduction of management measures and policy priorities. The red, blue, and yellow shading represent the periods I, II, and III, respectively. Except for the summer fishing moratorium, most of the measures were synchronous with other seas in China.

**Figure 3 biology-13-00226-f003:**
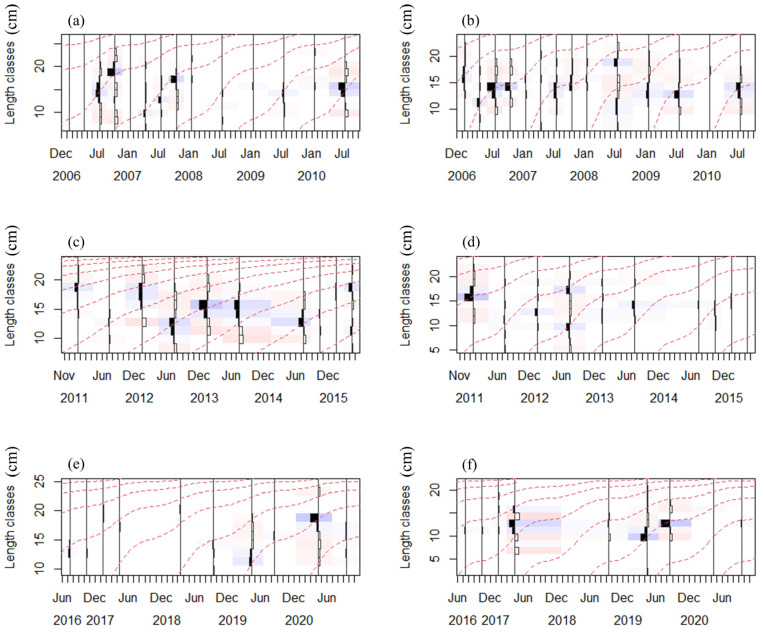
Raw length frequency data of the two commercial pelagic fishes from the Beibu Gulf, 2006–2020 (restructured data with a moving average setting of MA = 9, lines show estimated (red) growth curves plotted through the length frequency data); (**a**,**c**,**e**) represent the data of *D. maruadsi* in periods I, II, and III, respectively; (**b**,**d**,**f**) represent the data of *T. japonicus* in periods I, II, and III, respectively.

**Figure 4 biology-13-00226-f004:**
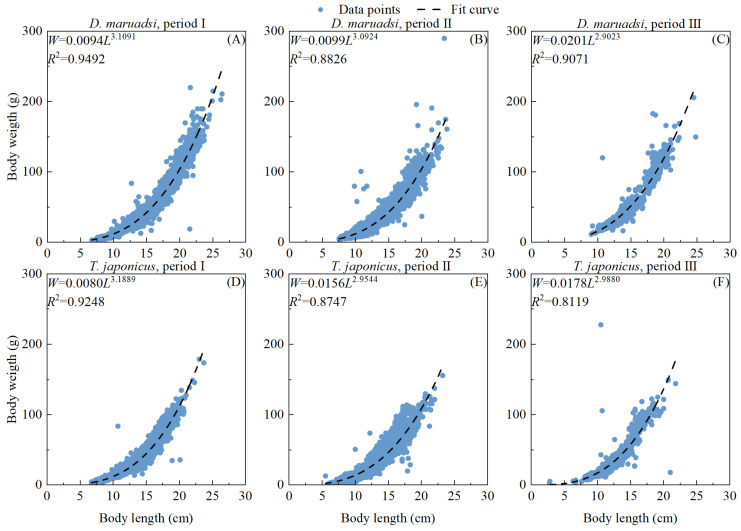
Size length–weight relationship of the two commercial pelagic fishes in the Beibu Gulf; (**A**–**C**) represent the data of *D. maruadsi* in periods I, II, and III, respectively; (**D**–**F**) represent the data of *T. japonicus* in periods I, II, and III, respectively.

**Figure 5 biology-13-00226-f005:**
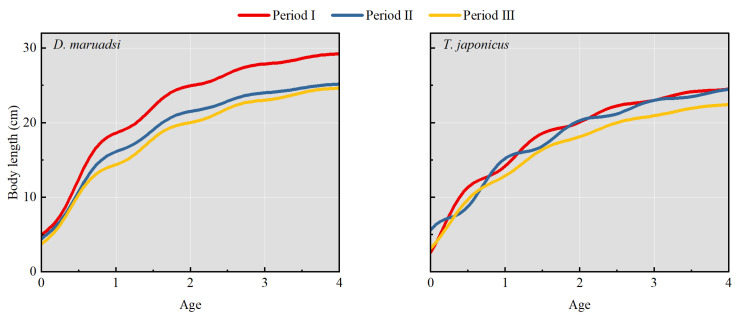
The seasonally oscillating growth curves of the two CPFs in the Beibu Gulf.

**Figure 6 biology-13-00226-f006:**
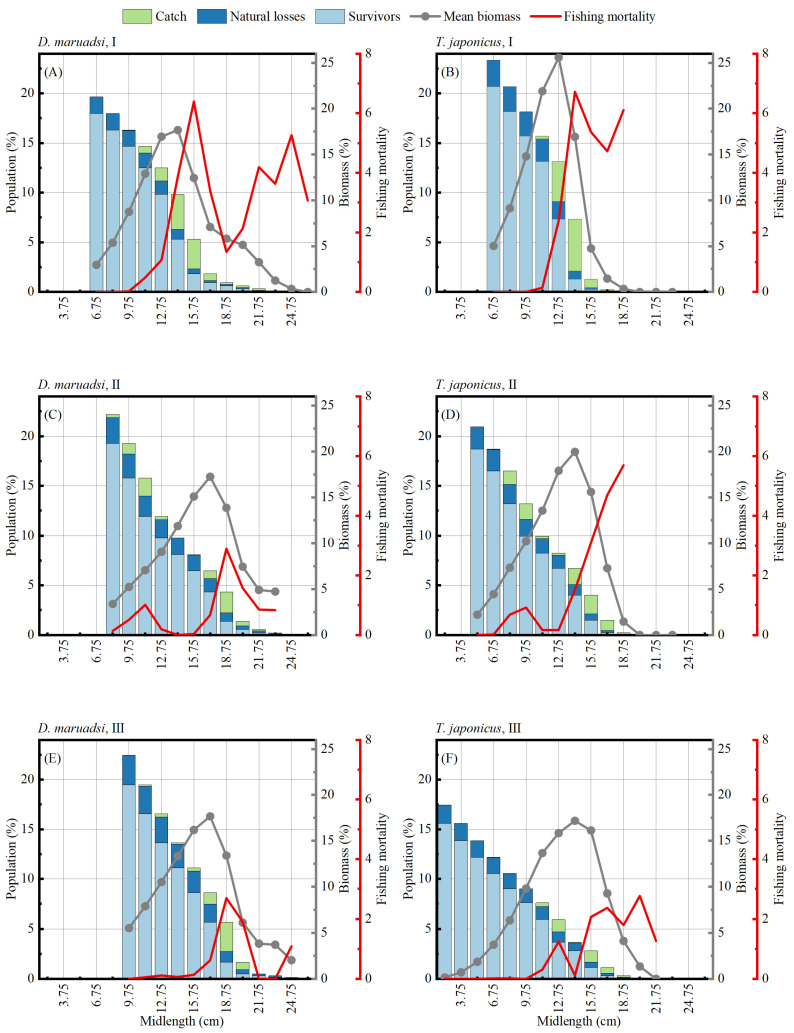
Results of Jones’ length-converted cohort analysis with reconstructed population structure. (**A**,**C**,**E**) represent the data of *D. maruadsi* in periods I, II, and III, respectively; (**B**,**D**,**F**) represent the data of *T. japonicus* in periods I, II, and III, respectively.

**Figure 7 biology-13-00226-f007:**
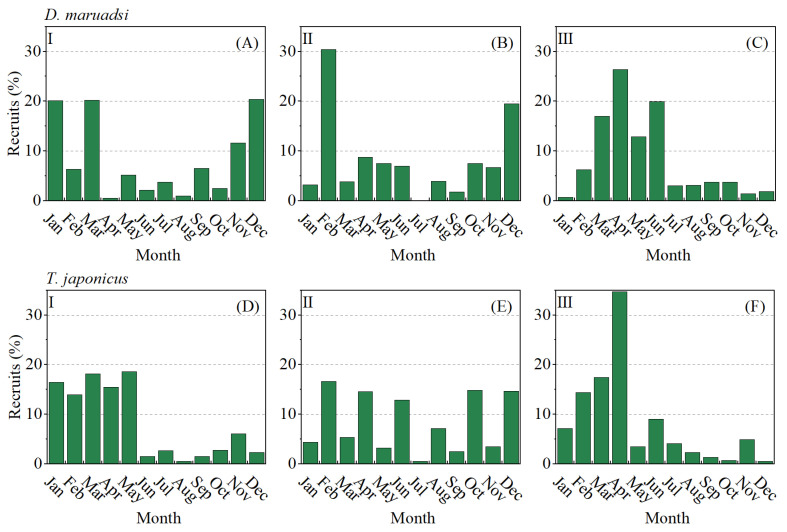
Recruitment pattern of the two pelagic fishes in the Beibu Gulf, estimated from the restructured length frequency data over an arbitrary 1-year timescale. (**A**–**C**) represent the data of *D. maruadsi* in periods I, II, and III, respectively; (**D**–**F**) represent the data of *T. japonicus* in periods I, II, and III, respectively.

**Figure 8 biology-13-00226-f008:**
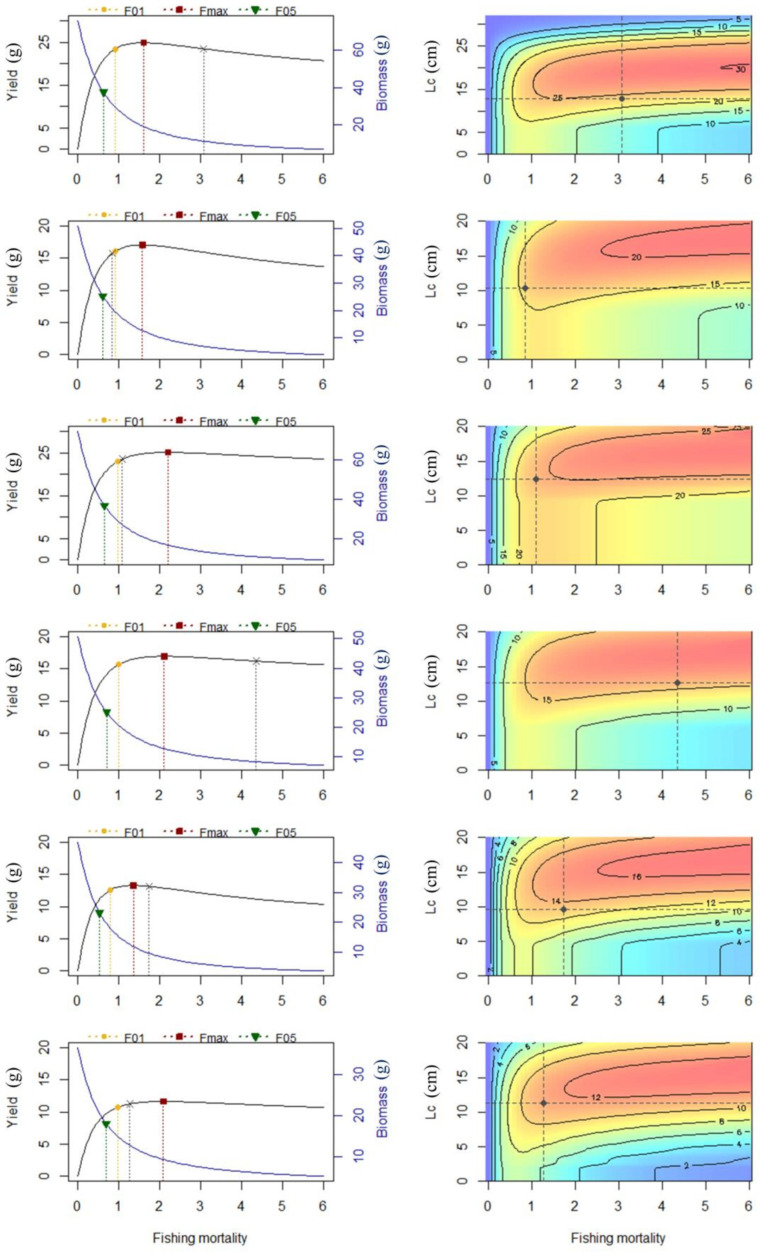
Results of the Thompson and Bell model: Column I—curves of yield and biomass per recruit; Column II—exploration of the impact of different exploitation rates and *L*_c_ values on the relative yield per recruit. F01, the fishing mortality rate where the slope of the yield per recruit curve is 10% of its value at the origin (*F*_0.1_); F05, the fishing mortality associated with a 50% reduction relative to the virgin biomass; Fmax, the mortality level when the yield is at a maximum (*F*_max_). The x-axis corresponds to the fishing mortality of the fully exploited length class(es). The black dots represent the current fishing regime. The x-axis corresponds to the fishing mortality.

**Table 1 biology-13-00226-t001:** Summary statistics of the size distribution of the two major commercial small pelagic fishes in the Beibu Gulf.

Species	Years	Periods	Number	*L_min_* (cm)	*L_max_* (cm)	*L_median_* (cm)	*L_mean_* (±SD) (cm)
*D. maruadsi*	2006–2010	I	4102	6.7	26.4	14.2	14.46 (±2.87)
	2011–2015	II	2475	7.5	23.8	14.2	14.41 (±2.78)
	2016–2020	III	701	9.0	24.8	13.6	14.32 (±3.07)
*T. japonicus*	2006–2010	I	6178	6.7	23.7	13.6	13.54 (±2.24)
	2011–2015	II	3872	5.5	23.2	13.0	13.01 (±2.58)
	2016–2020	III	1477	2.8	21.8	12.3	12.30 (±2.07)

Note: *L_min_* represents the minimum body length, *L_max_* represents the maximum body length, *L_median_* represents the median in body length, and *L_mean_* represents the mean body length.

**Table 2 biology-13-00226-t002:** VBGF parameter obtained for two CSPFs in the Beibu Gulf, 95% CI: 95% confidence intervals.

Species	Periods	*L_∞_* (cm) (95% CI)	*K* (a^−1^) (95% CI)	*C* (95% CI)	*t*_s_ (95% CI)	*φ*′ (95% CI)
*D. maruadsi*	I	**30.43** (23.94–32.69)	**0.77** (0.24–0.95)	**0.57** (0.32–0.88)	**0.54** (0.28–0.72)	**2.87** (2.33–2.94)
	II	**26.19** (19.52–28.04)	**0.77** (0.32–0.86)	**0.51** (0.22–0.74)	**0.55** (0.12–0.77)	**2.43** (2.39–2.74)
	III	**26.48** (22.70–28.89)	**0.63** (0.50–0.85)	**0.54** (0.22–0.90)	**0.46** (0.05–0.90)	**2.65** (2.59–2.72)
*T. japonicus*	I	**26.08** (24.18–27.82)	**0.68** (0.52–0.86)	**0.58** (0.32–0.85)	**0.23** (0.14–0.33)	**2.65** (2.59–2.74)
	II	**26.13** (19.64–28.86)	**0.63** (0.25–0.86)	**0.75** (0.29–0.91)	**0.75** (0.14–0.92)	**2.66** (2.31–2.82)
	III	**24.22** (22.06–26.66)	**0.62** (0.20–0.81)	**0.36** (0.17–0.89)	**0.32** (0.03–0.97)	**2.58** (2.08–2.69)

**Table 3 biology-13-00226-t003:** Mortality parameter, exploitation rates, and the selectivity of the two CPFs in the Beibu Gulf from 2006–2020.

Species	Periods	*Z*	*F*	*M*	*E*	*L*_50_ (cm)	*L*_75_ (cm)
*D. maruadsi*	I	4.17	3.07	1.10	0.74	12.82	13.48
	II	1.99	0.83	1.16	0.42	10.36	11.20
	III	2.08	1.08	1.00	0.52	12.35	12.64
*T. japonicus*	I	5.41	4.35	1.06	0.80	12.66	13.23
	II	2.74	1.74	1.00	0.63	9.63	10.19
	III	2.28	1.26	1.02	0.55	11.30	12.00

**Table 4 biology-13-00226-t004:** Effect of fishing mortality changes on biological reference points of the two CSPFs in the Beibu Gulf.

Species	Periods	*F* _0.1_	*F* _max_	*E* _max_	*YPR* _0.1_	*YPR* _max_	*YPR* _c_
*D. maruadsi*	I	0.91	1.60	0.38	23.44	24.87	24.89
	II	0.91	1.57	0.79	16.10	17.06	15.92
	III	0.98	2.19	1.05	23.05	25.08	23.50
*T. japonicus*	I	0.99	2.10	0.39	15.66	16.96	15.70
	II	0.79	1.35	0.49	12.60	13.31	12.73
	III	0.97	2.08	0.92	10.74	11.64	11.53

## Data Availability

The data used in this study were obtained from the literature, which are cited in the text and provided in the reference section.

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
