# Peer review of "Stock Assessment of the Commercial Small Pelagic Fishes in the Beibu Gulf, the South China Sea, 2006–2020"

_biology, 2024, doi:10.3390/biology13040226_

Round 1

Reviewer 1 Report

Comments and Suggestions for Authors

Good quality but only local significance. Suggestions for small corrections & improvements on the attached file.

Comments on the Quality of English Language

Minor corrections suggested. Some sentences too long and meaning has been lost (eg. sentence not finished).

Author Response

Dear reviewer:

Thank you for your letter and the reviewers’ comments on our manuscript entitled “Stock assessment of the commercial small pelagic fishes in the Beibu Gulf, the South China Sea, form 2006-2020” (Manuscript ID: biology-2867880). Those comments are very helpful for revising and improving our paper, as well as the important guiding significance to another research. We have studied the comments carefully and made corrections which we hope meet with approval. The main corrections are in the manuscript and the responds to the reviewers’ comments are as follows.

Specific comments

  1. processed the same way?

Response: Thank you for pointing out this problem in our manuscript. We have made corrections according to your request.

  1. recorded

Response: Thank you for your guidance. We have made corrections to this word.

  1. Not in a population but in the sample.

Response: Thank you for your reminder. We have changed this to "fish samples".

  1. More details advised of how the population level (~stock) was determined. There are many variations of the method(s) used.

Response: Thank you very much for your suggestion. We have added more introductions to terms such as Fmax and F01 in this section, enriching the application details of the method (line 220).

  1. Please fix all these.

Response: I'm very sorry for the reading inconvenience caused to you. The editorial department has informed me that this is due to a file format issue and it has been fixed.

  1. cm? Unit in the legend please

Response: Thank you for your reminder. We have revised and improved all the charts in this article accordingly.

286.

Response: Sorry for any inconvenience caused to your reading of this article. This is due to a file bug and has been fixed.

  1. due to the factors such as

Response: Thank you for pointing out this problem in our manuscript. We have made the corresponding corrections according to your requirements.

  1. decrease

Response: Your suggestion is very meaningful, but since "shrinking" is a noun used to describe the fixed combination of fish body size shrinking due to environmental changes, we will not make any modifications here

  1. Unclear. Please break into 2-3 shorter sentences.

Response: Sorry for the inconvenience caused to your reading due to our mistake. We have rearranged the sentence here according to your suggestion to make it easier for readers to understand.

  1. Same, sentences too long; meaning lost.

Response: Thank you for pointing out the sentence issues in this article. We have made comprehensive revisions to address them.

  1. Elaborate. There are ways to untangle fishing effects from environmental effects.

Response: Thank you for pointing out this problem in our manuscript. We have revised the wording here and added relevant references (line 383).

  1. simply put, decrease in fishing pressure

Response: Based on your suggestion, we have simplified the words in this section to make the expression more concise and fluent.

  1. Sentence not finished

Response: Thank you for correcting the errors in this section of the manuscript. We have revised it according to your request and added more content to make the expression more complete.

  1. serious abnormality

Response: Thank you for your suggestion. We have made corresponding corrections to this word.

  1. uniform

Response: Thank you for your rigorous review of the text in this manuscript. We have made corrections to this word.

  1. Do not generalize here - rather maintain a local flair for this paper (use Beibu Gulf; use your species; use this kind of the "reduced fishing pressure" you specifically had in your region).

Response: Thank you for pointing out this problem in our manuscript. We have rearranged the sentences in this section to better fit the main idea of the manuscript (line 449).

Reviewer 2 Report

Comments and Suggestions for Authors

Abstract, lines 22-23: The sentence “Laboratory-based analyses……sampling years” is not clear and difficult to understand.

Abstract, line 26: The 3 periods of study must be briefly detail in Abstract.

Abstract, line 35: How can the authors mentioned “positive” as all biological characteristics, in particular length-related indicators are getting worse.

Abstract, line 37: Climate change should not be mentioned since there is no related analysis in this manuscript.

Abstract, lines 38-42: Numerical results must be provided to support this claim.

Introduction, lines 59-60: Repetition with different cited references.

Introduction, lines 63-65: The 3 management policies must be elaborated

Introduction, lines 80-81: Better to elaborate “Obviously, previous fishery management… as a whole. Why they are not effective?

Introduction, lines 84-85: Too trivial and can be deleted.

Introduction, lines 115-117: The sentence is not clear.

Introduction, lines 124-130: Not clear on which parts of the study respond to (i) overiew of exploited statuses and (ii) effect of past fisheries management.

Materials and Methods, lines 137-138: Please elaborate “standardized fisheries resource surveys”

Materials and Methods, 2.2 sampling: If the 2 species are commercially targeted, why the samplings from commercial fisheries are not incorporated in the data.  By doing this, it would maximize and clear the modes of age groups in the length frequency data.

Figure 1 must be incorporated with scale bar. It is also seen from Figure 1 that the survey covered th lesser area from 2015, would this effect to the samples for length frequency data.?

Figure 2 and related text are not much clarified on how was the 3 period of study divided. Please elaborate more.

Materials and Methods, line 195: Very surprise me that these 2 fish species carapace!!!! As far as I know only boxfish that we can claim they have carapace.

Materials and Methods, line 201 and Eq. 5: Check the symbol of phi-prime.

Materials and Methods, line 216: The length-based yield per recruit model by Thompson and Bell (1934)??? No, it is Beverton and Holt’s Yield per Recruit model (1957)

Results, Growth parameters and growth curves: The growth curves must be incorporated to Figures 3 to see how they are well represented by the length frequency data.

Results, Mortality and selectivity: Surprising on the drastically dropped of F-values in Period II and III. Better show the graphical results on points those used for the length converted catch curves.

Results, Stock status: Better explain what did happened to the stock status by each monitored variables, in particular changes in F-array for the size classes.

Results, Recruitment and yield per recruit: The analysis of this recruitment pattern is not mentioned in M&M, as well is the recruitment pattern necessary?

Results, lines 302 – 312: Re-write the whole paragraph. The authors should go back the fundamental concept of YPR and explain the results accordingly.

Discussion: Not see clearly discuss on the drastic change of F-values

Discussion, lines 358-363: This sentence is difficult to follow and quite curious that whether the authors clear on the concept Rosa-Lee phenomenon since it is seemingly they took the first sentence in Abstract of Kraak et al (2019) to discuss. Better to re-write this sentence.

English of the manuscript must be extensively edited.

Captions for Figures and Tables must be revised and self-explained.

As a sampling from survey, the ethical approval for the study must be presented.

Comments on the Quality of English Language

Editing is required

Author Response

Dear reviewer:

Thank you for your letter and the reviewers’ comments on our manuscript entitled “Stock assessment of the commercial small pelagic fishes in the Beibu Gulf, the South China Sea, form 2006-2020” (Manuscript ID: biology-2867880). Those comments are very helpful for revising and improving our paper, as well as the important guiding significance to another research. We have studied the comments carefully and made corrections which we hope meet with approval. The main corrections are in the manuscript and the responds to the reviewers’ comments are as follows.

Specific comments

Abstract, lines 22-23: The sentence “Laboratory-based analyses……sampling years” is not clear and difficult to understand.

Response: Thank you for pointing out this problem in our manuscript. We have made corrections according to your request (line 33).

Abstract, line 26: The 3 periods of study must be briefly detail in Abstract.

Response: Thank you for your suggestion. We will provide a detailed explanation of the time periods for the three fisheries management periods in the abstract section as per your request (line 37).

Abstract, line 35: How can the authors mentioned “positive” as all biological characteristics, in particular length-related indicators are getting worse.

Response: Your reminder is very meaningful. Although the overall parameters of the two fish species in this study showed a deteriorating trend, there have been some improvements in recent times. Therefore, it is mentioned that fisheries management has a limited positive effect.

Abstract, line 37: Climate change should not be mentioned since there is no related analysis in this manuscript.

Response: Thank you for pointing out this problem in our manuscript. We have removed the relevant content on climate change in the abstract section (line 43).

Abstract, lines 38-42: Numerical results must be provided to support this claim.

Response: Thank you very much for your suggestion. We have made corresponding corrections to the sentences in this section (line 48-50).

Introduction, lines 59-60: Repetition with different cited references.

Response: Thank you for pointing out this problem in our manuscript. We apologize for this mistake and we have fixed it.

Introduction, lines 63-65: The 3 management policies must be elaborated

Response: Thank you for your reminder. We have provided a more detailed explanation and expression of this part of the fisheries management policy (line 71).

Introduction, lines 80-81: Better to elaborate “Obviously, previous fishery management… as a whole. Why they are not effective?

Response: Your opinion is very meaningful. As the introduction in the front of the manuscript has already sorted out and evaluated the history of fishery management in the South China Sea and the current situation of fishery resources in the area, this section is only a summary of specific conclusions and will not discuss the reasons for the high and low efficiency of fishery management.

Introduction, lines 84-85: Too trivial and can be deleted.

Response: Thank you very much for your strict review of the text in this manuscript. We have made the full text revisions according to your requirements (line 97).

Introduction, lines 115-117: The sentence is not clear.

Response: Sorry for the inconvenience caused to your reading. We have corrected this issue.

Introduction, lines 124-130: Not clear on which parts of the study respond to (i) overiew of exploited statuses and (ii) effect of past fisheries management.

Response: Your suggestion has greatly benefited us. We have made relevant revisions to address the issues in this section. In the first part, our main purpose is to analyze the status of fish populations. In the second part, our goal is to indirectly reflect the effectiveness of fisheries management based on the time series changes in population status (line 126).

Materials and Methods, lines 137-138: Please elaborate “standardized fisheries resource surveys”

Response: Thank you for your reminder. We have added complete relevant information in section 2.1 (line 142).

Materials and Methods, 2.2 sampling: If the 2 species are commercially targeted, why the samplings from commercial fisheries are not incorporated in the data.  By doing this, it would maximize and clear the modes of age groups in the length frequency data.

Response: Your suggestion is very insightful; however, this study did not consider the actual market caught fish data based on the following considerations: firstly, commercially caught fish have obvious selectivity, and there may be biases in judging the number of small individual fish; Secondly, the survey methods used in this study are relatively comprehensive and can provide as much exposure as possible to all the structures of the population (line 143).

Figure 1 must be incorporated with scale bar. It is also seen from Figure 1 that the survey covered th lesser area from 2015, would this effect to the samples for length frequency data.?

Response: Your suggestion is very practical. In addition, considering that the research subject of this study is upper middle level fish, which have high swimming ability, the reduction of the surveyed sea area to a certain extent will not have a significant impact on the research results (line 161).

Figure 2 and related text are not much clarified on how was the 3 period of study divided. Please elaborate more.

Response: Thank you for pointing out this problem in our manuscript. We have provided a detailed explanation of the characteristics of each fishery management period in Section 2.3.

Materials and Methods, line 195: Very surprise me that these 2 fish species carapace!!!! As far as I know only boxfish that we can claim they have carapace.

Response: Sorry, this was caused by our mistake. We apologize and have fixed this error (line 200).

Materials and Methods, line 201 and Eq. 5: Check the symbol of phi-prime.

Response: Thank you for your reminder. We have reviewed this again (line 207).

Materials and Methods, line 216: The length-based yield per recruit model by Thompson and Bell (1934)??? No, it is Beverton and Holt’s Yield per Recruit model (1957)

Response: Thank you for your reminder. We have made corrections to this section (line 219).

Results, Growth parameters and growth curves: The growth curves must be incorporated to Figures 3 to see how they are well represented by the length frequency data.

Response: Thank you for your suggestion. We have provided a more accurate description of Figure 3 according to the corresponding requirements (line 236).

Results, Mortality and selectivity: Surprising on the drastically dropped of F-values in Period II and III. Better show the graphical results on points those used for the length converted catch curves.

Response: Thank you very much for your suggestion. Your suggestion is very meaningful, and we will consider this content presentation issue with all authors.

Results, Stock status: Better explain what did happened to the stock status by each monitored variables, in particular changes in F-array for the size classes.

Response: Thank you for your guidance. We have provided more detailed information on this issue in the results section (line 286).

Results, Recruitment and yield per recruit: The analysis of this recruitment pattern is not mentioned in M&M, as well is the recruitment pattern necessary?

Response: Your suggestion is very meaningful. As the calculation of the recruitment pattern is the default output of the software, it is not explicitly mentioned in the methods section. The reason for this section is to consider the need to determine the possible changes in the recruitment pattern of the fish population.

Results, lines 302 – 312: Re-write the whole paragraph. The authors should go back the fundamental concept of YPR and explain the results accordingly.

Response: Thank you for your suggestion. We have rewritten the content of this section, supplemented necessary content, and reanalyzed the changes in YPR values (line 321).

Discussion: Not see clearly discuss on the drastic change of F-values

Response: Your suggestion is very meaningful, and we will re discuss the necessary F value in the discussion section based on your guidance (line 424).

Discussion, lines 358-363: This sentence is difficult to follow and quite curious that whether the authors clear on the concept Rosa-Lee phenomenon since it is seemingly they took the first sentence in Abstract of Kraak et al (2019) to discuss. Better to re-write this sentence.

Response: Thank you for pointing out the issue with the manuscript. We have revised the wording in this section to make it easier for readers to understand, emphasizing that individuals with faster growth rates may have larger body sizes compared to those with slower growth in the same generation, making them easier to capture (line 364).

English of the manuscript must be extensively edited.

Response: Thank you for your suggestion. We will revise the English sentences in this manuscript.

Captions for Figures and Tables must be revised and self-explained.

Response:Thank you for your reminder. We have revised all the charts.

As a sampling from survey, the ethical approval for the study must be presented.

Response: Thank you for your reminder. We will add relevant information at the end of the article.

Reviewer 3 Report

Comments and Suggestions for Authors

The article has practical significance for nature conservation and rational exploitation of aquatic biological resources. However, the scale of the study is so small and the objects are so specific that the work is unlikely to be of interest to a wide range of readers of the journal. If, nevertheless, the editors consider it possible to publish this manuscript, then you should pay attention to the following notes.

Key notes:

The authors forgot to write which part of the range of the species being studied is the sampling region. Is this the area of their permanent habitat, spawning or feeding migrations, the center or the outskirts of the range? The interpretation of the results depends on this.

To study small fish, the trawl must be equipped with a fine-mesh insert in the cod-end with a mesh of not more than 10 mm.

The manuscript does not say by what criterion the three periods I, II and III are distinguished. They do not correspond to the introduction of stock management measures, nor to the dynamics of catches, nor to changes in fishing effort.

The text does not say why the minimum length of one species increased over the years, while the second species, on the contrary, decreased (see Table 1).

All plots in Figure 4 show outliers that should be removed before performing regression analysis. These are errors in measuring the length or mass of individuals or errors in their species identification.

It remains unclear to what extent fishing has negatively affected the state of the two fish populations, and what the share of the influence of natural environmental factors is. It would be good to find literary information about changes in the size of fish of these species in parts of the range that are not exploited by fishing.

The article does not even attempt to compare the size of fish and other characteristics of their populations with interannual changes in temperature, salinity, oxygen, nutrients, abundance of food items, predators, parasites, etc.

Minor notes:

There is no need to enter the abbreviation "CSPFs". After it is written at the beginning that these are 2 species of fish, they are named in Latin, it is said that they are small and commercial, you can write simply fish. It will be clear to the reader what kind of fish you are talking about.

On Line 59-60, the phrase “due to a lack of restrictions on fishing yields” is repeated twice with different references.

Figure 1 is called "Map showing stations for bottom trawl surveys...", but not a single trawl station is shown. To do this, there must be 52 points on the map.

It is difficult to understand the meaning of the sentence on Lines 89-93. It needs to be rephrased and/or divided into several shorter sentences.

Figure 2 should show catch per effort, not catch and effort separately.

Line 195 mentions carapace length twice, but the fish studied do not have a carapace. What is it about?

The length of the fish was measured with an accuracy of 1 cm (see Line 152), the body length frequency was analyzed at 1.5 cm standard length intervals (Line 223 and Figure 3), but why then the average length of the fish in Table 1 is given with an accuracy of hundredths centimeter?

Reference must be verified. It does not, for example, contain Dunn et al., 2016, which is mentioned on Line 428.

Author Response

Dear reviewer:

Thank you for your letter and the reviewers’ comments on our manuscript entitled “Stock assessment of the commercial small pelagic fishes in the Beibu Gulf, the South China Sea, form 2006-2020” (Manuscript ID: biology-2867880). Those comments are very helpful for revising and improving our paper, as well as the important guiding significance to another research. We have studied the comments carefully and made corrections which we hope meet with approval. The main corrections are in the manuscript and the responds to the reviewers’ comments are as follows.

Specific comments

The authors forgot to write which part of the range of the species being studied is the sampling region. Is this the area of their permanent habitat, spawning or feeding migrations, the center or the outskirts of the range? The interpretation of the results depends on this.

Response: Thank you very much for your suggestion. Your suggestion is very meaningful. There are several reasons why this study did not specifically mention the specific ecological functions of fish in the study area. Firstly, the scope of the study area is relatively large, and the life cycle of fish in the area can be fully covered. Secondly, the research object of this study is fish with strong swimming ability, and they are mainly active around the study area.

To study small fish, the trawl must be equipped with a fine-mesh insert in the cod-end with a mesh of not more than 10 mm.

Response: Your suggestion is very meaningful. The data used in this study comes from fishery resource surveys, so the fishing targets are not only small fish. We believe that your suggestion should be given sufficient attention in future surveys.

The manuscript does not say by what criterion the three periods I, II and III are distinguished. They do not correspond to the introduction of stock management measures, nor to the dynamics of catches, nor to changes in fishing effort.

Response: Thank you very much for your feedback. This study divides fishery management periods based on the input and output of nearshore areas. We have supplemented the specific characteristics of each period in the manuscript (line 165).

The text does not say why the minimum length of one species increased over the years, while the second species, on the contrary, decreased (see Table 1).

Response: Thank you for your suggestions on the manuscript. Due to the human selection and randomness in sample collection, especially for the smallest individual body size, the variation of the smallest body size is not the focus of this study. The average body length and asymptotic body length are the most reflective values of population status.

All plots in Figure 4 show outliers that should be removed before performing regression analysis. These are errors in measuring the length or mass of individuals or errors in their species identification.

Response: Thank you very much for your reminder. In the regression analysis of this study, all outliers were automatically cleared by the software in advance during the iterative calculation. Therefore, the existence of outliers did not significantly affect the actual output results. The purpose of displaying outliers in the chart is to reflect the true situation of data recording.

It remains unclear to what extent fishing has negatively affected the state of the two fish populations, and what the share of the influence of natural environmental factors is. It would be good to find literary information about changes in the size of fish of these species in parts of the range that are not exploited by fishing.

Response: Your opinion is very constructive, however, a large number of studies in the past have shown that the fish resources in the region are being affected by overfishing, especially in terms of the response of fish size. The focus of this study is to focus on the changes in regional fishery resources over the past decade or five years, during which China's fishery management has made substantial progress.

The article does not even attempt to compare the size of fish and other characteristics of their populations with interannual changes in temperature, salinity, oxygen, nutrients, abundance of food items, predators, parasites, etc.

Response: Your guidance is very valuable, however, due to the limited space in this article, we are unable to delve into population dynamics in multiple aspects or fields. We apologize for any confusion this may have caused you in your reading.

There is no need to enter the abbreviation "CSPFs". After it is written at the beginning that these are 2 species of fish, they are named in Latin, it is said that they are small and commercial, you can write simply fish. It will be clear to the reader what kind of fish you are talking about.

Response: Thank you very much for your suggestion, but considering the unified characteristics of these two fish species, we believe that using abbreviations to refer to them is a common practice. We will also carefully consider your suggestion in future research.

On Line 59-60, the phrase “due to a lack of restrictions on fishing yields” is repeated twice with different references.

Response: Thank you for pointing out the errors in this part of the manuscript. We have corrected them (line 66).

Figure 1 is called "Map showing stations for bottom trawl surveys...", but not a single trawl station is shown. To do this, there must be 52 points on the map.

Response: Thank you very much for your reminder. We have added the location information of the relevant sites in Figure 1 (line 156).

It is difficult to understand the meaning of the sentence on Lines 89-93. It needs to be rephrased and/or divided into several shorter sentences.

Response: Thank you for pointing out the errors in this part of the manuscript. We have rewritten this section of the sentence (line 97).

Figure 2 should show catch per effort, not catch and effort separately.

Response: Your suggestion is very valuable; however, we mainly use government yearbook data for chart making and analysis. Therefore, to display specific fishery inputs and outputs, we did not use unit fishing effort as an indicator.

Line 195 mentions carapace length twice, but the fish studied do not have a carapace. What is it about?

Response: This is ours, and we apologize for any inconvenience caused to your reading.

The length of the fish was measured with an accuracy of 1 cm (see Line 152), the body length frequency was analyzed at 1.5 cm standard length intervals (Line 223 and Figure 3), but why then the average length of the fish in Table 1 is given with an accuracy of hundredths centimeter?

Response: Thank you very much for your suggestion. We have reviewed the manuscript again and corrected any errors in the unit identification.

Reference must be verified. It does not, for example, contain Dunn et al., 2016, which is mentioned on Line 428.

Response: Thank you very much for your suggestion. We have re-edited the references in the manuscript according to the requirements of the magazine.

Round 2

Reviewer 2 Report

Comments and Suggestions for Authors

Nothing in terms of scintific but another round of English editing should be done.

Comments on the Quality of English Language

Another round of English editing should be done.

Reviewer 3 Report

Comments and Suggestions for Authors

I understand that the authors decided to limit themselves to minimal editing of the manuscript, concentrating their efforts on responding to the reviewer. This answer has the information we need on many issues. It's a pity that the reader doesn't recognize it. However, the manuscript has been improved. I will dispense with further comments.